

# Aluminium and base cation chemistry in dynamic acidification models – need for a reappraisal?

Jon Petter Gustafsson[1,2], Salim Belyazid[3], Eric McGivney[1], Stefan Löfgren[4]

[1]Department of Sustainable Development, Environmental Science and Engineering, KTH Royal Institute of Technology, Teknikringen 10B, 100 44, Stockholm, Sweden.
[2]Department of Soil and Environment, Swedish University of Agricultural Sciences, P.O. Box 7014, SE-750 07 Uppsala, Sweden.
[3]Department of Physical Geography, Stockholm University, SE-106 91 Stockholm, Sweden
[4]Department of Aquatic Sciences and Assessment, Swedish University of Agricultural Sciences, P.O. Box 7050, SE-750 07 Uppsala, Sweden

**Correspondence**: Jon Petter Gustafsson (gustafjp@kth.se)

**Abstract.** Long-term simulations of the water composition in acid forest soils require that accurate descriptions of aluminium and base cation chemistry are used. Both weathering rates and soil nutrient availability depend on the concentrations of $Al^{3+}$, of H+, and of base cations ($Ca^{2+}$, $Mg^{2+}$, $Na^+$ and $K^+$). Consequently, assessments of the acidification status and base cation availability will depend on the model being used. Here we review in what ways different dynamic soil chemistry models describe the processes governing aluminium and base cation concentrations in the soil water. Furthermore, scenario simulations with the HD-MINTEQ are used to illustrate the difference between model approaches. The results show that all investigated models provide the same type of response to changes in input water chemistry. Still, for base cations we show that the differences in the magnitude of the response may be considerable depending on whether a cation-exchange equation (Gaines-Thomas, Gapon) or an organic complexation model is used. The former approach, which is used in many currently used models (e.g. MAGIC, ForSAFE), causes stronger pH-buffering over a relatively narrow pH range, as compared to state-of-the-art models relying on more advanced descriptions in which organic complexation is important (CHUM, HD-MINTEQ). As for aluminium, a fixed gibbsite constant, as used in MAGIC and ForSAFE, leads to slightly more pH-buffering than in the more advanced models that consider both organic complexation and $Al(OH)_3(s)$ precipitation, but in this case the effect is small. We conclude that the descriptions of acid-base chemistry and base cation binding in models such as MAGIC and ForSAFE are only likely to work satisfactorily in a narrow pH range. If the pH varies greatly over time, the use of modern organic complexation models is preferred over cation exchange equations.

## 1 Introduction

Acid rain has been an environmental issue of concern ever since Svante Odén presented his famous newspaper article in 1967, in which the acidification of water systems in the northern hemisphere was first described (Odén, 1967). Most of the



atmospheric acid deposition was caused by sulphur emissions from fossil fuel combustion. In Europe and North America successful efforts were finally made to reduce the emissions, and in 2014 the atmospheric sulphur emissions in Western Europe were less than 20 % of what they were around 1970, when the emissions peaked (Engardt et al. 2017). Despite the drastic cuts of emissions, there are still acidified soils and waters for which the critical loads are exceeded at the current atmospheric S

5 deposition level. In 1980, the critical loads were exceeded for 58 % of all Swedish lakes. In 2030, exceedances are expected for between 3 and 22 % of the lakes (Moldan et al. 2017). The exact figure depends mostly on the intensity of forest harvesting (Akselsson et al. 2007; Iwald et al. 2013). The latter leads to net removal of base cations, and is therefore also an acidifying process that affect the critical loads (Nilsson et al. 1982). Hence, problems with soil and water acidification could persist even though the atmospheric deposition is cut to very low levels.

For a long time dynamic acidification models have been valuable tools to understand the underlying mechanisms and to produce scenarios for long-term acidification effects. Most of the dynamic acidification models in use today have their roots in ideas and concepts developed in the 1970s and 1980s. For example, cation sorption was viewed as an exchange process where $Ca^{2+}$, $Mg^{2+}$, $Na^+$, $K^+$, $H^+$ and $Al^{3+}$ competed for a limited number of exchange sites. This view persisted despite the fact

that many of acidified sites had soils rich in organic matter but with a low clay percentage. In addition, the mechanisms responsible for Al dissolution were still poorly understood at the time.

It was not until the late 1980s that it was suggested that the Al solubility in many surface horizons was controlled by organic matter complexation (Mulder et al. 1989), and a few years later it was shown that the pH-dependent dissolution of Al deviated

significantly from the cubic relationship expected from equilibrium with an $Al(OH)_3$-type phase (Mulder and Stein, 1994; Berggren and Mulder, 1995; Wesselink and Mulder, 1995). Moreover, with the development of organic complexation models such as WHAM, it was shown that the solubility of base cations was better described when the variable-charge nature of organic matter was taken into account (Tipping and Hurley, 1992; Tipping et al. 1995). Since then, modern organic complexation models such as WHAM, NICA-Donnan (Kinniburgh et al. 1999) and SHM (Gustafsson, 2001) have been applied

to numerous soil systems, and the results showed the correctness of the above conclusions (e.g. Lofts et al. 2001; Weng et al. 2002; Gustafsson and Kleja, 2005)

Despite these findings, many of the most used dynamic acidification models such as ForSAFE (Warfvinge et al. 1993; Wallman et al. 2005), MAGIC (Cosby et al. 1985; Cosby et al. 2001) and SMART-VSD (de Vries et al. 1989; Posch and Reinds, 2009)

are not updated with these more modern process descriptions. In the following, we refer to these models as 'ion-exchange' models, as the solid-solution interactions are based primarily on ion-exchange equations. However, some dynamic models have been developed that contain updated descriptions. These are CHUM-AM (Tipping, 1996; Tipping et al. 2006), which uses WHAM, and SMARTml (Bonten et al. 2011), which includes the NICA-Donnan model. Recently, the HD-MINTEQ



model was added to this list (Löfgren et al. 2017). HD-MINTEQ uses the SHM. However, the latter models have (at least so far) been less used in acidification research.

The purpose of this paper is to review the process descriptions of currently used models as regards base cations and aluminium, and then to investigate the difference in model performance between the two types of model mentioned above, i.e. between ion-exchange models and organic complexation models. This is done through scenario simulations in which the atmospheric deposition of S and sea salt is varied in steps. The nature of the modelled response is compared. The results suggest that the difference in model performance is rather small as long as the pH remains within a confined pH range; the most significant differences are seen for proton buffering and for base cation dynamics.

## 2 Aluminium and base cations in dynamic acidification models – Review

### 2.1 Aluminium and base cations in ion-exchange models

In MAGIC, SAFE/ForSAFE, as well as in the SMART/VSD models, a "gibbsite" equilibrium, i.e. a cubic relationship between the $Al^{3+}$ and $H^+$ activities, $\{Al^{3+}\}$ and $\{H^+\}$, is used to calculate $\{Al^{3+}\}$ from pH, according to:

$$K_G = \frac{\{Al^{3+}\}}{\{H^+\}^3} \tag{1}$$

In the models, the $K_G$ value is treated as a fitting parameter that is optimised from measurements/estimates of pH and $\{Al^{3+}\}$ in a given soil horizon. In other words, there is not necessarily any *a priori* assumption involved that states than an $Al(OH)_3(s)$ phase governs $\{Al^{3+}\}$. Rather, equation 1 should be seen as a practical way of accounting for Al solubility, which may hold if pH and $\{Al^{3+}\}$ do not vary much over time (i.e., so that the ion activity product of $Al(OH)_3(s)$ remains constant).

Sorption (and desorption) of base cations is treated as an exchange process in the traditional models, where cations compete for a constant number of exchange sites. In MAGIC (Cosby et al. 1985; Cosby et al. 2001), the Gaines-Thomas cation exchange formalism is used. Four equations are used for describing the exchange of $Al^{3+}$ with $Ca^{2+}$, $Mg^{2+}$, $Na^+$ and $K^+$. In the case of Al-Ca exchange, the equation reads:

$$K_{GT, Al/Ca} = \frac{E_{Ca}{}^3 \{Al^{3+}\}^2}{E_{Al}{}^2 \{Ca^{2+}\}^3} \tag{2}$$





where $K_{GT,Al/Ca}$ is the selectivity coefficient for Al/Ca exchange, and $E_{Al}$ and $E_{Ca}$ are the equivalent fractions of exchangeable $Al^{3+}$ and $Ca^{2+}$, respectively. The $Al(OH)_3(s)$ equilibrium (c.f. above) is used to fix $\{Al^{3+}\}$. Further, in MAGIC the activities of exchangeable $H^+$ and of other exchangeable ions (e.g. $Fe^{3+}$, $Mn^{2+}$) are not considered. According to Cosby et al. (2001) this is because these are assumed to be insignificant. Hence the cation exchange capacity (CEC), i.e. the total number of exchange

sites, is considered to be the sum of exchangeable base cations and exchangeable $Al^{3+}$. The sum of exchangeable fractions is 1;

$$E_{Al} + E_{Ca} + E_{Mg} + E_{Na} + E_K = 1 \qquad\qquad (3)$$

During calibration of the MAGIC model, it is usually assumed that the CEC can be estimated using a conventional extraction of exchangeable acidity (e.g. 1 M KCl). By using data for exchangeable base cations (e.g. by 1 M $NH_4Cl$) the equivalent fractions of all cations can be estimated. If solution data are available or can be estimated, Eq. (2) can then be used to calculate selectivity coefficients. The optimised coefficients should be regarded as site-specific, i.e. they cannot be generalized to other soils or even to other soil layers. This has been identified as a weak point of the ion-exchange models when comparing them

to more process-oriented models such as CHUM-AM and SMARTml, for which site-specific optimisation of any constants or selectivity coefficients should not be necessary (Bonten et al. 2011).

In the SAFE/ForSAFE suite of models (Warfvinge et al. 1993; Wallman et al. 2005), as well as in the SMART and VSD models (de Vries et al. 1989; Posch and Reinds, 2009), the Gapon ion-exchange equation is used instead of the Gaines-Thomas

equation. Moreover, in the standard versions of these models the base cations $Ca^{2+}$, $Mg^{2+}$ and $K^+$ are bulked into a single entity, $BC^{2+}$, where $K^+$ is treated as a divalent ion whereas $Na^+$ is usually not considered at all. As a consequence, no more than three exchange equations need to be considered, which describe Al/BC, H/BC, and H/Al exchange. For example, in the case of Al/BC exchange, the Gapon equation reads:

$$K_{G,AlBC} = \frac{E_{BC}\{Al^{3+}\}^{1/3}}{E_{Al}\{BC^{2+}\}^{1/2}} \qquad\qquad (4)$$

The number of equations required can be further reduced to two through the gibbsite equilibrium, which fixes $\{Al^{3+}\}$ as a function of $\{H^+\}$ (Warfvinge et al. 1993; Posch and Reinds, 2009). The calibration procedure of the SMART/VSD and the SAFE/ForSAFE models is similar to the one for MAGIC, i.e. measured exchangeable fractions (after assuming that the total

number of sites is obtained from exchangeable acidity) are used to calculate values for the selectivity coefficients.

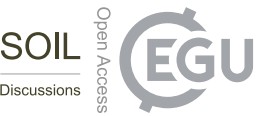

In all models, the pH of the soil solution is calculated from charge balance, which makes it important to correctly estimate the charge contributions from organic acids and Al-organic complexes. This is done in slightly different ways in the ion-exchange models. The SAFE/ForSAFE and SMART-VSD models use the Oliver equation (Oliver et al. 1983) to estimate the organic anion charge. Although the Oliver equation incorporates the effect of Al-organic complexes on the calculated organic anion

5 charge, Al-organic complexes are not considered in the mass balance of the organic anion component. To consider the effect on the mass balance of Al, the ForSAFE/SAFE models use the following empirical equation, originally suggested by Tipping et al. (1988):

$$[Al_{org}] = \alpha \cdot [DOC] \cdot [Al^{3+}]^{\beta} \cdot [H^+]^{\gamma} \tag{5}$$

where $[Al_{org}]$ is organically bound Al, $[DOC]$ is dissolved organic carbon (mg L$^{-1}$), $\alpha = 2.09 \cdot 10^{-7}$, $\beta = 0.718$, $\gamma = -1.054$. The concentration unit for $[Al_{org}]$, $[Al^{3+}]$ and $[H^+]$ is mol L$^{-1}$.

MAGIC uses a triprotic acid model to estimate the organic anion charge (Cosby et al. 2001). This model considers the effect

of Al-organic complexation by formation of two complexes between $Al^{3+}$ and the organic ligand. While MAGIC is normally set up using just a single box representing a soil profile, the SAFE/ForSAFE models use three or four soil layers in sequence, which may represent different horizons of a soil profile.

### 2.2 Aluminium and base cations in CHUM-AM and in HD-MINTEQ

Tipping (1996) was the first to present a dynamic acidification model (CHUM) based on the concept of organic matter complexation as an important process for Al and base cation chemistry in the surface horizon of soils. In this model, WHAM (Tipping and Hurley, 1992; Tipping, 1994) was connected to submodels accounting for parameters such as water flow, micropore-macropore exchange and reactions involving nitrogen. The model was designed to run on a daily time step, and considered temporal differences in precipitation, water contents etc. Mineral weathering was considered with a simple power

law. CHUM also included a submodel for predicting DOC in the soil solution, using the fulvic acid adsorption/desorption model of Tipping and Woof (1991). In the original version of CHUM, the sorption and desorption of base cations are assumed to only occur through electrostatic interactions and to a lesser extent (for $Ca^{2+}$ and $Mg^{2+}$) through complexation to a variably charged humic surface according to the humic ion-binding Model V in WHAM.

Later, Tipping et al. (2006) presented an updated version of CHUM called CHUM-AM, in which Model VI (Tipping, 1998) was used instead. This model has mostly been used to simulate the long-term behaviour of metals in UK and in Switzerland (Tipping et al. 2010; Rieder et al. 2014). In one paper, however, the CHUM-AM model was used to simulate the changes in



water chemical composition of three Cumbrian lakes over a period of 1000 years (Tipping and Chaplow, 2012). The simulated values of pH and dissolved major ions were in reasonable agreement with measurements made during the last 30-40 years.

The recently introduced HD-MINTEQ model (Löfgren et al. 2017) is intended for scenario simulations of pH and dissolved

ions in forest soils using a weekly time step. In the study of Löfgren et al. (2017), HD-MINTEQ was used to predict the soil chemical response to a stepwise change in S deposition, with and without whole-tree harvesting. HD-MINTEQ employs the Stockholm Humic Model (Gustafsson, 2001) to estimate the binding of base cations and aluminium to the soil solid phase. $SO_4$ adsorption is considered using the extended Freundlich equation of Gustafsson et al. (2015).  Nitrogen chemistry is not explicitly simulated, instead the user supplies the concentration of dissolved $NO_3^-$ and $NH_4^+$ in the dissolved phase as input to

the model. The current version of HD-MINTEQ incorporates the PROFILE weathering model (Sverdrup and Warfvinge, 1993), for details see McGivney et al. (2018).

CHUM-AM and HD-MINTEQ share many similarities in how they treat the reactions relevant for aluminium and base cations. In both models, the acid dissociation of humic and fulvic acid determines the amount of negative charge as well as the extent

of pH buffering. The dissociation of a carboxylic or a phenolic acid group is defined as follows:

$$ROH \rightleftarrows RO^- + H^+, \qquad\qquad K_i \qquad\qquad\qquad (6)$$

, where R represents the humic molecule. Here, $K_i$ is an intrinsic dissociation constant, which is defined as:

$$K_i = \frac{\{RO^-\}\{H^+\}}{\{ROH\}} \cdot \omega \qquad\qquad\qquad (7)$$

, where $\omega$ is an electrostatic correction term, which is calculated differently in Model VI and SHM (Tipping, 1998; Gustafsson, 2001).  Both models employ a discrete-site approach to describe the pH dependence of the dissociation of humic substances

(HS). There are 8 ROH sites of different acid strengths and so there are 8 $K_i$ values. The 4 most strongly acid sites ($i = 1$-4) are referred to as type A sites, whereas sites 5-8 are type B sites. The most strongly acid sites probably represent mainly carboxylic acid groups, whereas the type B sites are thought to represent weaker acids such as phenolic acids. Four constants (log $K_A$, log $K_B$, $\Delta pK_A$ and $\Delta pK_B$) are needed to define the 8 log $K_i$ values, according to:

$i = 1$-4   log $K_i$ = log $K_A$ - $\Delta pK_A$ $\qquad\qquad\qquad\qquad\qquad$ (8)

$i = 5$-8   log $K_i$ = log $K_B$ - $\Delta pK_B$ $\qquad\qquad\qquad\qquad\qquad$ (9)




The total number of proton-dissociating sites, $n$ (mol g$^{-1}$) is the sum of all type A and B sites. Within each site group (A or B) all sites are present in equal amounts. For fulvic acids, the total amount of type B sites is 30 % (SHM) or 50 % (Model VI) of the amount of type A sites, whereas for humic acids, the amount of type B sites is 50 % of that of the type A sites in both models.

Cations and metals may be sorbed to the proton-dissociating sites through both electrostatic attraction and complexation, according to equations given elsewhere (Tipping, 1998; Gustafsson, 2001). For weakly sorbing ions such as $Ca^{2+}$, $Mg^{2+}$ and $K^+$, electrostatic attraction will typically be the predominating mechanism in acid forest soils, whereas for $Al^{3+}$ and $Fe^{3+}$, complexation will be more important (Tipping, 2002). In Model VI / CHUM-AM, a mixture of mono-, bi- and tridentate

complexes is allowed to form between the metals and the organic ligands, whereas in the case of SHM, the complexation of $Al^{3+}$ and $Fe^{3+}$ is exclusively through the formation of bidentate complexes. In both models, the formation of $Al(OH)_3(s)$ and $Fe(OH)_3(s)$ is allowed when the solubility constants are exceeded, using the solubility data of Gustafsson et al. (2001) and Liu and Millero (1999), respectively.

**2.3 Aluminium and base cations in SMARTml**

The SMARTml model combines the SMART soil acidification model (de Vries et al. 1989) with a scaled-down version of the VSD model (Posch and Reinds, 2009), and employs Orchestra (Meeussen, 2003) for chemical equilibrium calculations. This includes cation binding to organic matter using the NICA-Donnan model (Kinniburgh et al. 1999). Similar to CHUM/AM and HD-MINTEQ, SMARTml assumes organic matter to be the predominant sorbent phase for base cations and aluminium. In

addition, the use of the NICA-Donnan model implies that HS dissociation governs both proton buffering and surface charge properties. In the model, the negative charge of the humic molecules gives rise to attraction of counterions, which accumulate in the vicinity of the molecules, inside a Donnan phase, the volume of which ($V_D$) is given by an empirical relationship. All of the charge on the humic particle, $q$ (mol$_c$ kg$^{-1}$), is assumed to be completely neutralized by counterions within the Donnan volume. This leads to the following charge balance expression:

$$\frac{q}{V_D} + \sum_i z_i \left( c_{D,i} - c_i \right) = 0 \tag{10}$$

, where $c_{D,i}$ is the concentration of component $i$ with charge $z_i$ in the Donnan volume, and $c_i$ is its concentration in the bulk solution (which is generally much smaller). Furthermore, the Donnan model assumes a relationship between $c_{D,i}$ and $c_i$:

$$c_{D,i} = \chi^{z_i} \cdot c_i \tag{11}$$



, where $\chi$ is a Boltzmann term with an implicit Donnan potential; however, the latter does not need to be calculated in order to solve for $Q_i$ and $c_{D,i}$ in Eqs. (10) and (11). The amount bound of component $i$ onto one site is given by the following equation:

$$5 \qquad Q_i = \frac{n_i}{n_{ref}} \cdot Q_{max,ref} \cdot \frac{\left(\widetilde{K}_i \cdot c_{D,i}\right)^{n_i}}{\sum_i \left(\widetilde{K}_i \cdot c_{D,i}\right)^{n_i}} \cdot \frac{\left\{\sum_i \left(\widetilde{K}_i \cdot c_{D,i}\right)^{n_i}\right\}^p}{1 + \left\{\sum_i \left(\widetilde{K}_i \cdot c_{D,i}\right)^{n_i}\right\}^p} \qquad (12)$$

, where $Q_i$ is the amount bound of component $i$, $n_i$ is the component-specific non-ideality parameter, $n_{ref}$ is the non-ideality parameter for the reference species (usually $H^+$), $Q_{max,ref}$ is the maximum binding capacity for the reference species, $\tilde{K}_i$ is the median value of the affinity distribution for species $i$, and $p$ is the width of the distribution. The NICA-Donnan equation is

10 applied for two sites, considering a carboxylic ("weak") and a phenolic ("strong") site, each with its own parameters, but with a common Donnan phase. Usually, the NICA-Donnan model is used for humic and fulvic acids separately.

For aluminium, SMARTml uses the same basic approach as CHUM-AM and HD-MINTEQ, i.e. it assumes Al solubility to be governed by organic complexation as long as the soil solution is undersaturated with respect to $Al(OH)_3(s)$. If not, equilibrium

with respect to $Al(OH)_3(s)$ is assumed (Bonten et al. 2011). The solubility of base cations is governed mainly by electrostatic attraction to the Donnan phase of the soil humic and fulvic acid.

So far, the SMARTml model has been applied only for one site, the Solling Forest, Germany, for which pH, major ions and trace elements of the soil solutions were simulated (Bonten et al. 2011; Bonten et al. 2015).

**3 Methods**

**3.1 Comparison of model descriptions – general approach**

Due to the different model descriptions of base cation and aluminium chemistry, and the importance that these reactions have on the overall acid-base chemistry of forest soils, we hypothesized that modelled dissolved base cations and aluminium dynamics would be substantially different depending on the model setups and inputs. However, for purposeful comparisons, a

25 number of considerations need to be made. First, to specifically study the differences in base cation and aluminium dynamics, other model-specific differences (e.g., how the models handle plant uptake, N chemistry, $SO_4$ adsorption, etc.) need to be eliminated. Second, comparisons of "real-world" scenarios are difficult to design as the input drivers (e.g., deposition) are changing continuously. For these reasons, we chose to set up the comparison as follows:





• Four models, representing different model descriptions, were defined: the 'ion-exchange A', 'ion-exchange B', 'SHM' and 'SHM with fixed Al(OH)$_3$' models. These are described in detail below.

• To avoid software-specific differences, all models were set up in the HD-MINTEQ model framework (Löfgren et al.
2017).

• The models were set up with four sequential soil layers in which the water was infiltrated from the soil surface through the O, E, B1 and B2 horizon, with steadily reduced rates accounting for evapotranspiration. The same general soil parameters were used for all models (Table 1). The parameter values were based on observations from the Gårdsjön site in SW Sweden (c.f. McGivney et al. 2018).

• Rather than using constant deposition values, the effects of sudden changes in deposition were simulated. This enabled a comparison of how quickly and in what ways the models responded to a deposition change. The deposition values for the first 20 years represent the estimated deposition in 1880 at the Gårdsjön site (Table 2), which may be seen as typical "preindustrial" values. The deposition during the following 40 years was taken from deposition chemistry measurements in 1980 (Löfgren et al., 2011), whereas for the period between year 60 and year 120, the projected deposition for 2020 was used.
We used the method of Ferm and Hultberg (1995) for estimates of dry deposition.

• To account for soil mineral weathering, the PROFILE model was used (Sverdrup and Warfvinge, 1988). The mineralogical composition shown in Table 1 was based on the mineralogy given by Sverdrup et al. (1998) for the Gårdsjön site. The mineral surface area was estimated from particle-size analysis using the relationship of Sverdrup (1996).

• SO$_4$ adsorption was considered using the Freundlich-based model of Gustafsson et al. (2015). As no SO$_4$ adsorption
data were available for this soil, it was assumed that 'some' adsorption occurred in the B horizons, using the parameter values for the Tärnsjö Bs soil studied by Gustafsson et al. (2015).

• For simplicity, the loss of base cations from the system due to plant uptake and harvest was set to zero throughout the simulation.

• In all simulations, dissolved organic C (DOC), NH$_4^+$ and NO$_3^-$ were set at constant values throughout the simulation;
the values given in Table 1 represent average values of these parameters for the period 1996-2011, based on soil lysimeter data from the Gårdsjön site (Löfgren et al. 2011).

## 3.2 The models – brief description and initialization

*Ion-exchange model.* In the ion-exchange model, both acid-base and cation binding reactions were conceptualized as ion-exchange reactions. The Gaines-Thomas equation, which is available as an option in the calculation core of HD-MINTEQ,
was used for these reactions. Two model variants were used, termed models A and B. In ion-exchange model A, the Gaines-Thomas equation was implemented as in MAGIC (described above and in Cosby et al. 2001), which did not specifically consider H$^+$ exchange. However, because this model had issues concerning the O horizon (c.f. Results section), a second model (B) was introduced, which also considered Al/H exchange. To consider the organic anion charge from dissolved organic carbon




(DOC), the triprotic acid model of Hruška et al. (2003) was used. Further, Eq. (5) of Tipping et al. (1988) was employed to estimate organically bound Al. To get the equivalent fractions of cations, we used results from $NH_4Cl$ extractions of exchangeable base cations and from KCl extractions of exchangeable Al and acidity, performed during 2010 at the Gårdsjön site within the Integrated Monitoring programme (Löfgren et al. 2011). Water chemistry from soil lysimeters was also available

from this site (Löfgren et al. 2011). With these data it was possible to calculate the selectivity coefficients for each layer given in Table 1. The log $*K_s$ value for $Al(OH)_3(s)$, needed to fix dissolved $Al^{3+}$ in the model, were also constrained from the 2010 lysimeter data. To be consistent with previous applications of the ion-exchange models, the log $*K_s$ value for $Al(OH)_3(s)$ was assumed not to be temperature-dependent.

*SHM.* For the simulations in which the organic complexation model SHM was used for metal binding (Gustafsson, 2001; Gustafsson and Kleja, 2005), we used the same assumptions as detailed by Löfgren et al. (2017). Briefly, it was assumed that 30 % (O horizon) or 50 % (other layers) of the soil organic matter was active with respect to proton and cation binding. Of the active organic matter in the O horizon, 75 % was assumed to be humic acid, whereas 25 % was attributed to fulvic acid. For the other horizons, these percentages were 50 % and 50 %, respectively. This results in the humic and fulvic acid concentrations

shown in Table 1. Further it was assumed that all dissolved organic matter (DOM) was geochemically active, and was assumed to be fulvic acid. Geochemically active Al is a fitting parameter in HD-MINTEQ (Table 1). It provides the total pool of Al in the model, and its value was kept constant during the simulation. The value of geochemically active Al was constrained initially by running the model using historic data for deposition (McGivney et al. 2018), and comparing the model result with lysimeter data. The value was chosen that provided the best description of both pH and Al combined.

*SHM with fixed $Al(OH)_3$.* The purpose of this model was to investigate the effect of having the Al concentration fixed by an $Al(OH)_3(s)$ equilibrium constant rather than letting the Al solubility be calculated from organic complexation equilibria in case the conditions were undersaturated with respect to $Al(OH)_3(s)$. In all other respects, the model was identical to the 'pure' SHM. This model was used to investigate to what extent it matters whether a sophisticated organic complexation model is used to

calculate Al solubility, or whether a 'gibbsite' constant is sufficient. The $Al(OH)_3(s)$ equilibrium constants were calculated in the same way as in the ion-exchange model.

*Model initialization.* A set of initial guesses were made of total suspension concentrations (adsorbed + dissolved) of base cations and $SO_4$. The models were run for at least 1,000 years with the 1880 deposition data to obtain steady-state conditions,

and then applied to the scenarios defined below.



### 3.3 Scenarios considered

Two scenarios were designed in HD-MINTEQ to investigate the effect of sudden changes in the input deposition (Table 2). This should give information on how the models handle these disturbances, and what the role of different cation binding models are.

• *Background-acid-background.* Year 0-20: 1880 deposition; year 20-60: 1980 deposition; year 60-120: projected 2020 deposition.

• *Background-salt.* In this scenario there was a sudden increase in the seasalt deposition after 40 years, and this increased deposition remained until the end of the simulation. The results were compared to a scenario in which the 1880

deposition remained throughout the simulation.

To further illustrate the differences between the cation binding models used, the outcome of a titration with $Ca(OH)_2$ of the O horizon was carried out. This was done in Visual MINTEQ 3.1 (Gustafsson, 2018) using identical parameters as in the HD-MINTEQ model simulation.

**4 Results and Discussion**

### 4.1 Response to changes in acid deposition

Selected results from the *Background-Acid-Background* simulation are shown in Fig. 1, Fig. 2 and Fig. 3. All models were able to provide simulations of pH and dissolved Ca that, at first sight, may seem reasonable. However, for the O horizon ion-exchange model A behaved differently from the other three models in that the response of the pH and of dissolved Ca was

20 much faster as the acid deposition changed. In addition, the concentration of exchangeable Ca barely changed at all, while clear responses were seen for the other models (Fig. 3). This can be attributed to the omission of exchangeable $H^+$ in ion-exchange model A. As a result, the only constituent that could change in response to the pH change was Al, which was governed by the $Al(OH)_3(s)$ equilibrium.

However, because the concentration of free $Al^{3+}$ ions in the O horizon was low (in the order of $2 \times 10^{-8}$ mol $L^{-1}$) the amount of $Al^{3+}$ exchanged for $Ca^{2+}$ during each time step was very small, causing an extremely slow response in soil water chemistry. By introducing exchangeable $H^+$, the ion-exchange model (B) behaved differently and responded to the changes in input chemistry similar to the SHM models. According to the latter three models, for the O horizon it would take between 10 and 20 years after a change to arrive within 0.05 pH units from the new steady-state pH value, with the longest times during the

recovery phase.





The above results suggest that it is necessary to include $H^+$ exchange in dynamic ion-exchange models if organic horizons are being included as separate layers. At present this is done in ForSAFE but not in MAGIC; however, in current practice, MAGIC is set up only for one soil layer where there is, on average, much higher dissolved $Al^{3+}$.

For the subsoil horizons the simulated response times were longer, in particular for the B horizons. In the latter, adsorption/desorption reactions involving both cations and $SO_4$ were important. As $SO_4$ adsorption was treated identically in all models, the differences observed are due to the different cation binding models used.

The ion-exchange models and the organic complexation models predicted similar trends in pH and Ca responses to changed

acid input. However, there were considerable differences in the magnitude and the exact timing of the changes. For example, the amplitude of pH variations in the E and B1 horizons was larger for the SHM models than for the organic complexation models (Fig. 1). The clearest model-specific differences were observed for the simulated values of exchangeable $Ca^{2+}$ (Fig. 3), for which the ion-exchange models predicted between 1.5 and 4 times higher values for the preindustrial amount of exchangeable $Ca^{2+}$ for the O, E and B1 horizons. As a consequence, according to the SHM the stores of exchangeable $Ca^{2+}$

were nearly emptied in the upper soil horizons due to the acid input, while higher values were maintained in the ion-exchange models.

**4.2 Response to changes in seasalt input**

The SHM and ion-exchange model B models, being the most sophisticated variants of the two types of model, were compared in terms of their behaviour in the *Background-salt* scenario. In qualitative terms, both models showed similar responses to a

sudden increase of the seasalt input (Fig. 4). An initial dip in pH was followed by a slow rebound, with the fastest response times observed for the O horizon, whereas for the B horizons the results indicate that decades or more are required to reach a new steady state. In the models, different mechanisms were responsible for this pattern. In ion-exchange model B, increased levels of $Na^+$ and $Mg^{2+}$ from the seasalt caused immediate ion exchange with $H^+$ and $Al^{3+}$, with a decreased solution pH as a consequence. With time however, the exchange proceeded until a new steady-state was established, which was close to the old

steady-state pH. By contrast, for the SHM, the increased ionic strength of the deposition input changed the dissociation properties of soil organic matter (SOM), so that a larger fraction of SOM was dissociated at a given pH (c.f. Gustafsson and Kleja, 2005); this led to the dissolution of $H^+$ and consequently the pH was decreased. In other words, in the SHM ion exchange was not the driving mechanism, although the results bore a superficial resemblance to those expected from an ion-exchange process. Again, a new steady-state pH, being closer to the original pH, was then slowly established with time.

As Fig. 4 shows, there were subtle but rather clear differences in how the two models simulated the temporal development of pH. First, for SHM the new steady-state pH values differed more from the original ones. This can be attributed to the larger significance of ion activity coefficients in the reactions for organic complexation when compared to the Gaines-Thomas



exchange equations in ion exchange model B. Second, the rebound of pH towards a new steady-state was slower in the case of SHM. The reason for this is not entirely clear but appears to be the result of a complex interplay of several differences between the models.

### 4.3 Isolating the pH effect – how well do the models describe a base titration?

To illustrate the differences in how the models handle a pH change, we used Visual MINTEQ 3.1 (Gustafsson, 2018) to simulate a titration in which $Ca(OH)_2(s)$ (which was assumed to dissolve completely) was added in steps to the Gårdsjön O horizon. The latter was assumed to have the chemical composition from 1880, except that all $Ca^{2+}$ was removed before the first titration step. The model simulations were made using ion-exchange model B and the two SHM models.

Although the two SHM models (with and without 'fixed' gibbsite constant) performed very similarly there were considerable differences between the SHM models and ion-exchange model B (Fig. 5). For the SHM, the soil buffered the input of $Ca(OH)_2(s)$ over a wide pH range, reflecting the wide distribution of $pK_a$ values of SOM (Gustafsson, 2001), while for the ion exchange model, the buffering occurred over a relatively narrow pH range, during which the buffer curve was very steep. The results also illustrate the difference in the number of cation-binding sites: for the O horizon, the optimised SHM models

contained about six times as many sites. However, most of these were not accessible to $Ca^{2+}$ except at very high pH. By contrast for ion-exchange model B, $Ca^{2+}$ occupied 70 % of the exchange sites already at pH 4.5 (Fig. 5, right panel).

### 4.4 Implications

The simple exercise shown in Fig. 5 illustrates a fundamental difference in model behaviour, a difference that should be important when discussing any model-specific differences observed in the scenarios. For example, the consistently higher

levels of preindustrial ion-exchanged $Ca^{2+}$ that the ion-exchange models predict can be considered a logical consequence of the procedure to calibrate ion-exchange coefficients from present-day observations and apply the model to describe preindustrial conditions when the pH was higher. This is caused by the poor ability of the ion-exchange models to describe the acid-base chemistry of the main cation sorbent in soils (= organic matter) properly.

Despite this, the overall impression of the scenarios shown in Figs. 1–4 is that the differences in model performance between the two types of model were less dramatic than might have been expected, when the ion-exchange equations were integrated with other processes in a dynamic model. Overall the results do not support the view that the state-of-the-art models for organic complexation will necessarily produce much more accurate dynamic modelling results compared to the ion-exchange equations and fixed gibbsite constants currently used in the ForSAFE and MAGIC suite of models. Other factors such as uncertainties

in deposition and uptake values, calibration procedures, etc. are likely to be of larger significance for the overall model performance.

Even so, however, there are several strong arguments for replacing the ion-exchange models with state-of-the-art organic complexation models in dynamic ecosystem models. One such argument is that the number of optimized coefficients for each simulated layer is lower for the organic complexation models (see e.g. Table 1). Second, the empirical basis for the organic complexation models is much stronger (see, e.g. Tipping, 2002; Gustafsson and Kleja, 2005); in other words, we know from

laboratory experiments that the latter are able to describe pH buffering and cation binding well in most acidic soils. Third, the results from the $Ca(OH)_2(s)$ titration, as well as the high preindustrial values for exchangeable $Ca^{2+}$ in the surface layers for the ion-exchange models, illustrate that the ion-exchange equations currently used in ForSAFE and MAGIC are only likely to work satisfactorily within a relatively confined pH range. If large pH fluctuations occur, the inability of the ion-exchange models to consider the variable-charge properties of the sorbent can cause substantial errors in the results. As an example, this

may impact the acidification classification of Swedish streams and lakes, for which MAGIC is used for estimating the pH difference between 1860 and currently. Moldan et al. (2013) estimated this pH difference to be smaller than -0.4 pH units in approximately 800 of 2903 lakes in 2030.

*Data and code availability.* Data on soil parameters at Gårdsjön are available from the Integrated Monitoring website at

http://info1.ma.slu.se/im/IMeng.html. The HD-MINTEQ code can be made available upon request to gustafjp@kth.se.

*Competing interests.* The authors declare that they have no conflict of interest.

*Special issue statement.* This article is part of the special issue "Quantifying weathering rates for sustainable forestry"

(BG/SOIL inter-journal SI). It is not associated with a conference.

*Acknowledgments.* This study was funded by the Swedish Research Council Formas (reg. no. 2011-1691) within the strong research environment "Quantifying weathering rates for sustainable forestry (QWARTS)". Partial funding to the simulations of the effect of seasalt input was provided by the Acidification programme of SLU's Environmental Monitoring and

Assessment.

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



**Table 1.** Parameter values and assumptions

| | O horizon | E horizon | B1 horizon | B2 horizon |
|---|---|---|---|---|
| Soil layer thickness (cm) | 5 | 5 | 20 | 15 |
| Bulk density (kg m$^{-3}$) | 156 | 773 | 749 | 836 |
| Annual discharge (m$^3$ m$^{-2}$) | 0.55 | 0.5 | 0.45 | 0.45 |
| Soil moisture (m$^3$ m$^{-3}$) | 0.3 | 0.3 | 0.3 | 0.3 |
| Temperature (ºC) | 8 | 8 | 8 | 8 |
| Organic C (g kg$^{-1}$) | 400 | 50 | 60 | 50 |
| SO$_4$ adsorption | None | none | some[a] | some[a] |
| DOC (mg L$^{-1}$) | 35 | 12.6 | 9.8 | 9.8 |
| Dissolved NO$_3^-$ (µmol L$^{-1}$) | 0.5 | 0.4 | 0.4 | 0.4 |
| Dissolved NH$_4^+$ (µmol L$^{-1}$) | 0.5 | 2.6 | 4.2 | 4.2 |
| CO$_2$ pressure (atm) | $1 \times 10^{-3}$ | $2 \times 10^{-3}$ | $7 \times 10^{-3}$ | $1 \times 10^{-2}$ |
| log *$K_s$, Al(OH)$_3$(s)[b] | -4.2 | -7.7 | -9.4 | -9.4 |
| *PROFILE parameters* | | | | |
| Mineral surface area (m$^2$ g$^{-1}$) | 0 | 1.2 | 1.1 | 2.0 |
| K-feldspar (%) | 0 | 15 | 18 | 19 |
| Plagioclase (%) | 0 | 14 | 15 | 16 |
| Hornblende (%) | 0 | 0.5 | 1.5 | 1.5 |
| Epidote (%) | 0 | 0.5 | 0.75 | 1.0 |
| Garnet (%) | 0 | 0.1 | 0.1 | 0.1 |
| Biotite (%) | 0 | 0.5 | 0.5 | 0.5 |
| Chlorite (%) | 0 | 0.4 | 0.4 | 0.4 |
| Vermiculite (%) | 0 | 3.0 | 15 | 5.0 |
| Apatite (%) | 0 | 0.1 | 0.2 | 0.3 |
| *Parameters specific for ion-exchange model[c]* | | | | |
| CEC (cmol$_c$ kg$^{-1}$) | 18.7 / 23.9 | 3.62 / 4.85 | 4.89 / 6.49 | 4.73 / 5.61 |
| log $K_{GT,Al/Ca}$ | -5.15 / -5.04 | -3.50 / -3.38 | -1.54 / -1.42 | 0.13 / 0.20 |
| log $K_{GT,Al/Mg}$ | -3.50 / -2.43 | -1.11 / -0.99 | 0.48 / 0.60 | 2.31 / 2.38 |
| log $K_{GT,Al/Na}$ | -1.54 / -1.33 | -1.00 / -0.74 | -0.35 / -0.10 | 0.35 / 0.50 |
| log $K_{GT,Al/K}$ | -6.41 / -6.20 | -6.32 / -6.07 | -4.72 / -4.47 | -4.11 / -3.96 |
| log $K_{GT,Al/H}$ | - / -4.80 | - / -6.76 | - / -7.73 | - / -7.11 |
| *Parameters specific for SHM* | | | | |
| Active HA (g kg$^{-1}$) | 180 | 25 | 22.5 | 22.5 |
| Active FA (g kg$^{-1}$) | 60 | 25 | 22.5 | 22.5 |
| Geochemically active Al (mmol kg$^{-1}$) | 40 | 50 | 80 | 80 |

[a]'Some' SO$_4$ adsorption – parameters for the Tärnsjö Bs1 soil (Gustafsson et al., 2015)
[b]Only used in simulations in which the solubility of Al(OH)$_3$(s) was set at a constant value
[c]The first value is for ion-exchange model A, and the second is for ion-exchange model B.



**Table 2.** Total deposition values used in the scenarios (mmol m$^{-2}$ yr$^{-1}$)

| Constituent | Background-acid-background | | | Background-salt | |
|---|---|---|---|---|---|
| | Year 0-20 | Year 20-60 | Year 60-120 | Year 0-40 | Year 40-120 |
| $SO_4^{2-}$ | 12.1 | 89.5 | 12.4 | 12.1 | 18.0 |
| $Ca^{2+}$ | 9.2 | 9.2 | 9.2 | 9.2 | 11.3 |
| $Mg^{2+}$ | 17.5 | 17.5 | 17.5 | 17.5 | 28.5 |
| $K^+$ | 9.5 | 9.5 | 9.5 | 9.5 | 11.6 |
| $Na^+$ | 166 | 166 | 166 | 166 | 264 |
| $Cl^-$ | 201 | 201 | 201 | 201 | 315 |



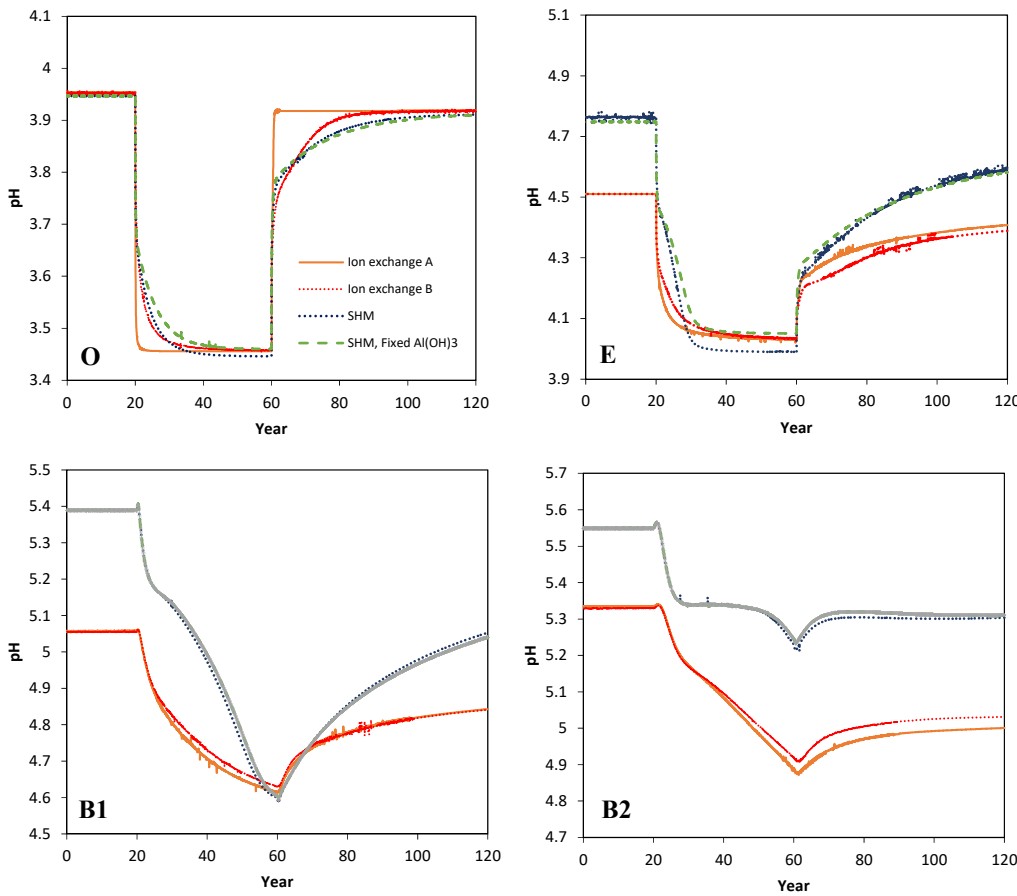

**Figure 1.** Simulated pH values as a function of time for the four models considered in the *Background-acid-background* scenario. For model setup and scenario parameters, see text.





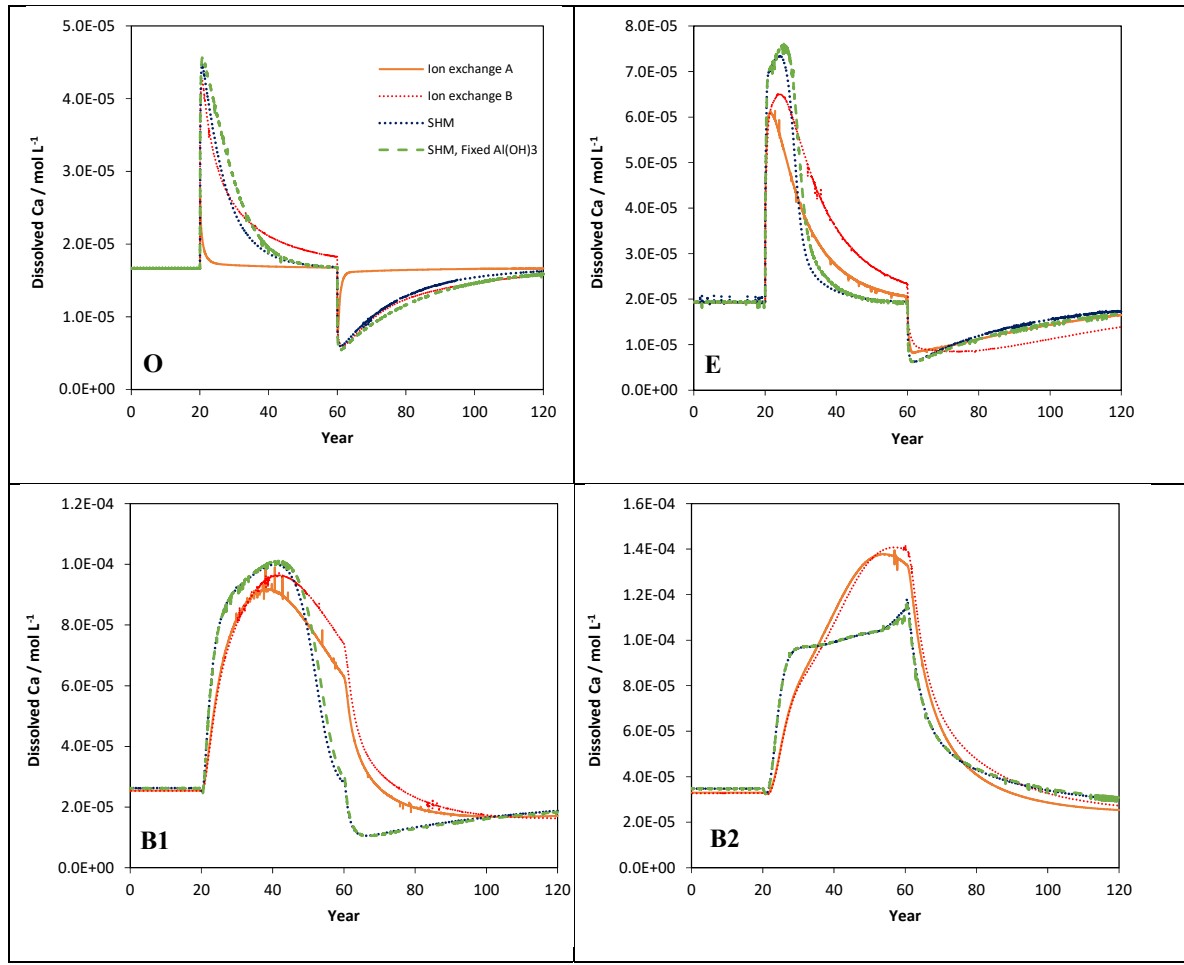

**Figure 2.** Simulated values of dissolved Ca as a function of time for the four models considered in the *Background-acid-background* scenario. For model setup and scenario parameters, see text.





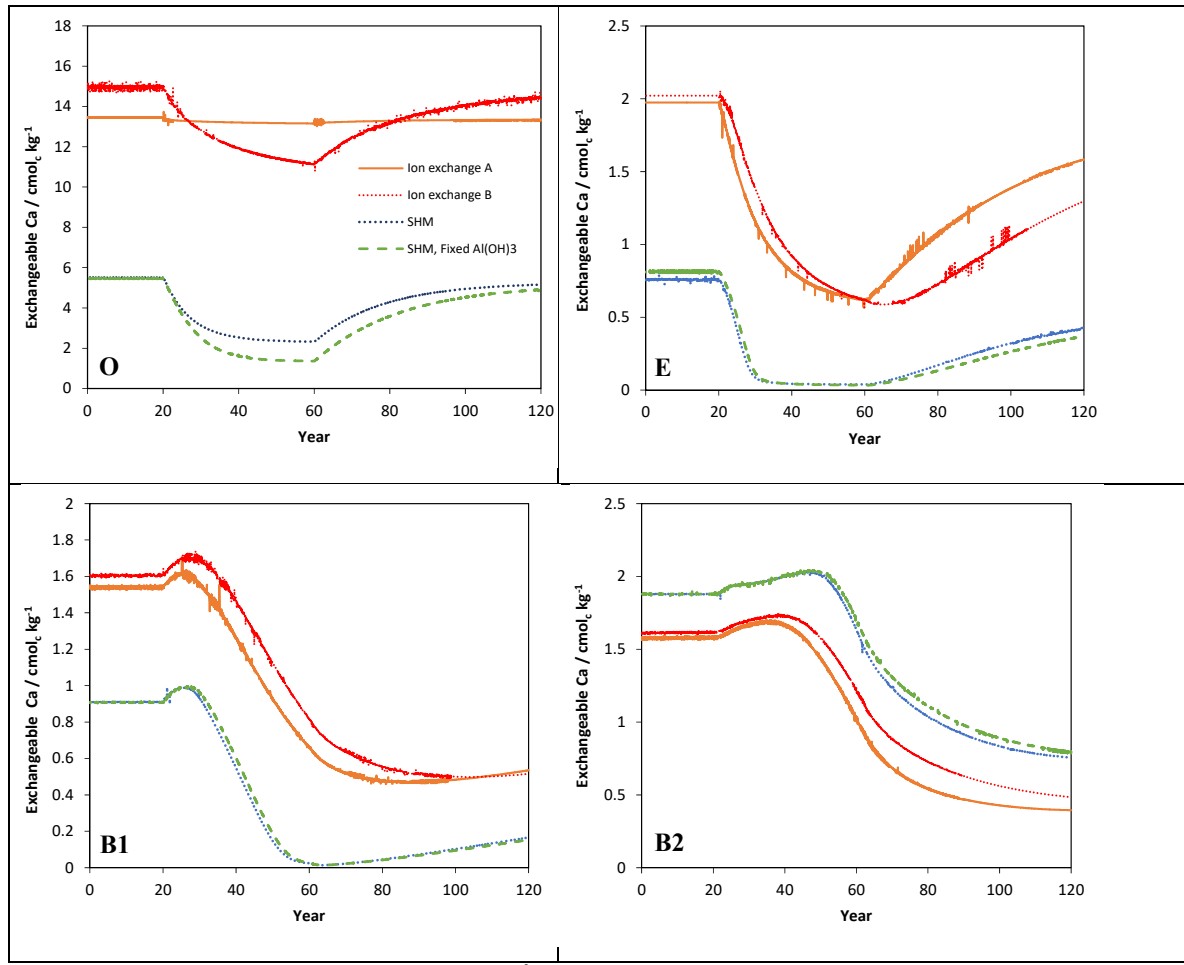

**Figure 3.** Simulated values of exchangeable (sorbed) $Ca^{2+}$ as a function of time for the four models considered in the *Background-acid-background* scenario. For model setup and scenario parameters, see text.



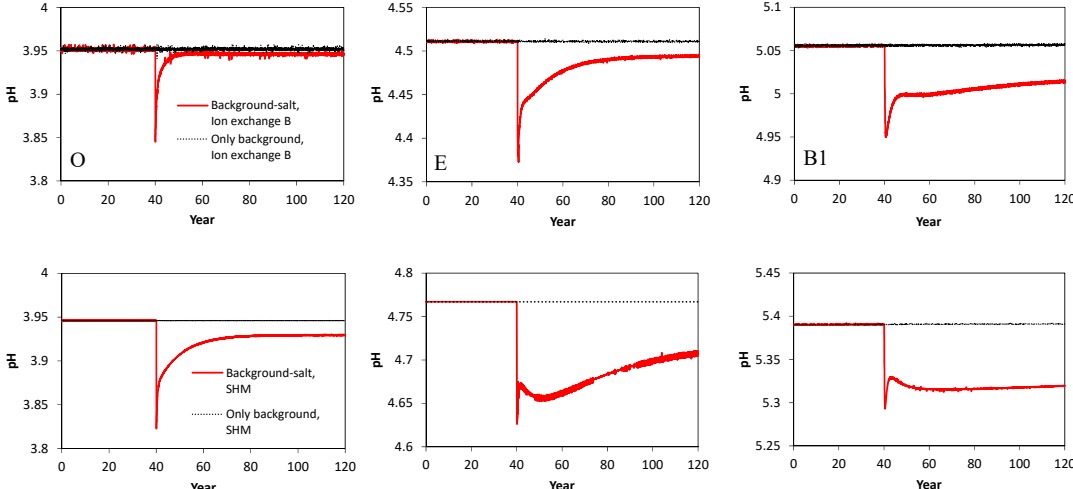

**Figure 4.** Simulated pH values following a permanent increase of the seasalt deposition after 40 years. The atmospheric deposition during years 0 to 40 (to 120 for the control simulation) was the estimated one at 1880 for the Gårdsjön site. Upper row: results obtained with the ion exchange model B, lower row: results obtained with the SHM.





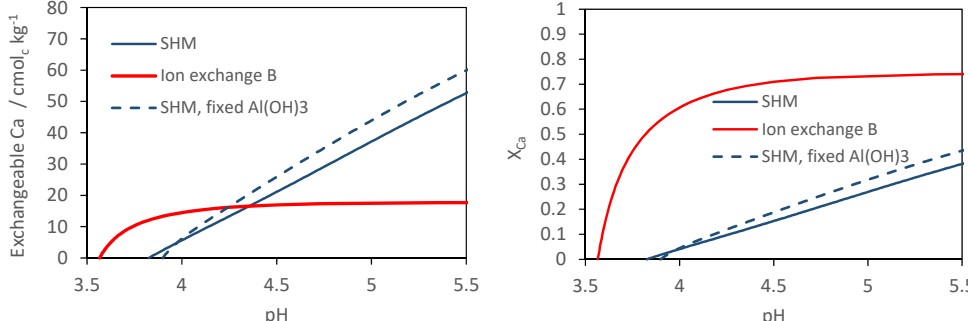

**Figure 5.** Model-simulated Ca(OH)$_2$ titration of the Gårdsjön O horizon with three different models. Left panel: exchangeable Ca$^{2+}$ as a function of pH; right panel: $X_{Ca}$ (the fraction of exchangeable Ca$^{2+}$ to the total number of binding sites) as a function of pH.