# Peer review of "Aluminium and base cation chemistry in dynamic acidification models – need for a reappraisal?"

_SOIL, 2018_

## Referee Comment (RC1) · Anonymous Referee #1 · 3 Aug 2018

Review of MS SOIL-2018-19:

**"Aluminium and base cation chemistry in dynamic acidification models = need for a reappraisal?"** by J.P. Gustafsson *et al.*

submitted for publication in *SOIL*.

General remarks:

This relatively short paper makes a case for replacing cation exchange modelling by organic complexation modelling in dynamic acidification models. They shortly describe the two approaches and then make their case by comparing the output of four different model systems. This is done well (although sometimes very concise), except the description (equations) of the models, which in two cases are quite deficient (for details see below); but that should not be too difficult to fix.
Overall, I consider the paper suitable for publication, after the authors have also taken into consideration the (sometimes minor) remarks/corrections listed below.

Detailed remarks:
*Note: 'X → Y' means: replace 'X' by 'Y' (in the text).*

**Abstract**:
P[age]1, L[ine]15 H+ → $H^+$.
P1, L15: Delete 'Consequently, ': The sentence is not a real consequence of the previous one (it's always true),
P1, L18: Insert 'model' after 'HD-MINTEQ'.

**Introduction:**
P3, L1: Insert 'model as a basis' (or 'as a basis') after 'SHM'.

**Aluminium and base cations in dynamic acidification models – a review:**
P4, L18: It could/should be mentioned that in SMART/VSD the user can choose between a Gapon or Gaines-Thomas exchange model.
P5, L31: Insert 'the' before 'UK'.
P5, L32: Why 'however' after 'In one paper'? Suggest deleting!
P6, L17: What does the stand-alone '$K_i$' in eq.6 mean? In the next line it is referred to as 'Here the $K_i$ is …'?
P6, L30: **Eq.8 does not make sense**: If I insert $i$=1,2,3,4 into it, I get 4 times the identical value for $\log K_1$, …, $\log K_4$, since the right-hand side of eq.8 does not depend on the index $i$!! Correct!
P6, L31: Same again!!
P8, L2: **There is no $Q_i$ in equations (10) or (11), thus one cannot solve for them!**!

**Methods**:
P11, L6: It is not essential, but the scenario name is a bit 'strange'; and, actually slightly misleading, since the 3rd interval is not background depo (but close).
P11, L8: Also this name is not 'unconfusing' …
Why not just call the 'Acid' and 'Salt'? – just a thought …
P11, L13: Should it be '(Gustafsson, 201**6**)', as in the References; or is the year there wrong?

**Results and Discussion:**
P11, L29: 'longest' → 'longer'.
P12, L2: Also in SMART/VSD $H^+$ exchange is modelled!
P13, L5: '(Gustafsson, 201**6**)'?
P13, L29: 'SMART/VSD' could be added (?)

P14, L12: The ending is a bit abrupt …

**References:**
Insert 'and' before the last author in at least 3 cases: P15, L18; P16, L8; P18, L9.
P 15, L24: Is it Gustafsson 2016 or 2018?

**Tables:**
Table 1: Caption: Maybe expand text?
Write 'humic acids' and 'fulvic acids' instead of 'HA' and 'FA', resp., in the first column.

Table 2: Add 'scenario' after '*Background-acid-background*' and after '*Background-salt*' in the first line if the Table.

**Figures:**
Figure 1: Caption: (a) Insert 'in the four soil layers (O, E, B1, B2)' before 'as a function'; (b) insert 'deposition' before 'scenario'.
The dotted red line is not visible in the graphs as such! Why not use solid lines with 4 (sufficiently) different colours?

Figure 2: same as Fig.1

Figure 3: same as Fig.1

Figure 5: For 'aesthetic reasons' maybe interchange in the legends the lines 'SHM' and 'Ion exchange B' (twice) (?)

---

## Referee Comment (RC2) · Anonymous Referee #2 · 3 Aug 2018

Major remark This paper addresses a very relevant topic related to the role of organic complexation in view of soil acidification modelling. This has been discussed for more than several decades, but still not commonly used.

This paper clearly fits within the scope of SOIL. However, it is a pity that it only includes hypothetical scenario analysis illustrating the differences between the two concepts (ion exchange' versus 'organic complexation' models). Overall, the paper is a rather technical description various types of modelling approaches. This makes the paper less relevant for a broader audience. More important, however, a comparison with observations (such as pH, Al concentration and base saturation) is missing. Such a

comparison is absolutely needed to judge the performance of both concepts. I realize that this is not an easy task, but a validation/application of the presented modelling concept by using e.g. the Gårdsjön observations or observations from Wesselink and Mulder (1995) and/or Bonten et al. (2001) must be doable.

I therefore conclude that, although the current manuscript addresses a relevant issue and is well written, a major revision is required because a comparison with observations is missing.

Please also note the supplement to this comment:
https://www.soil-discuss.net/soil-2018-19/soil-2018-19-RC2-supplement.pdf

---

## Author Response (AR1)

First we would like to thank the reviewers and the topical editor for very good comments, which helped us to improve the quality of the manuscript. Below we provide a point-by-point response to the comments of, first, the topical editor, and then, the reviewers.

Topical editor

*'As you can see from the comments posted by the two reviewers, both see merit in your paper and consider it a valuable contribution to our understanding of acidification models. However, both reviewers also raise several points both with respect to the equations presented (reviewer 1) and to the lack of external validation of the model outcomes (reviewer 2).*

*Both issues will need to be addressed before the manuscript can be published. In your response to the reviewer comments regarding the equations, you indicate in my view adequate changes and explanation to address the issue. Please incorporate these in your revised manuscript. With respect to the lack of validation, I see your point in the focus lying on a review of performance of modelling under hypothetical scenarios where sharp changes can be applied, and appreciate that validation would not be easy to add. However, you must very clearly explain in the text of the manuscript why this choice was made.'*

We have provided a detailed response to the reviewer comments below. As concerns the last comment, "with respect to validation…" etc, we have tried to explain our choice more clearly by:

(i) including the following text in connection to the aims of the manuscript (p. 3, lines 6-10):

"Based on parameter values from the Gårdsjön integrated monitoring site in SW Sweden, this is done through hypothetical scenario simulations in which the atmospheric deposition of S and sea salt is varied in steps. Compared with using complex "real-world" data, this approach makes it possible to isolate the effect of the different model descriptions from other processes that affect soil chemistry (see Methods). Thus, the nature of the modelled response is compared and no attempt is made to "validate" a certain model description."

(ii) adding explanatory text also in the Methods section, p. 9, line 30 ff:

"An alternative way to compare the model descriptions would have been to use "real-world" data concerning e.g. deposition and plant uptake for a well-studied catchment such as Gårdsjön or Solling. However, such a comparison is more difficult to design purposefully as the input drivers (e.g., deposition) change continuously, so that a steady-state in, e.g. base cation stores is never obtained. In addition, other factors affecting the overall chemical response, e.g. N transformations and long-term differences in the processes affecting the organic C balance (litterfall, decomposition etc), would obscure the comparison. Moreover, model initialization is always an issue, due to the lack of data on

soil and water chemistry under preindustrial conditions – this could affect the comparison between different model descriptions as well, since the back-calculation of the initial ecosystem state would depend on the exchange and solubility models used. Hence, to isolate the effect of the different descriptions of base cations and aluminium from other processes that affect soil chemistry, and to "validate" a certain model description, would be a very challenging task. Therefore, although not directly relevant for field conditions, the hypothetical scenarios as presented here provide some clues that would not be readily obtained under the inherently complex conditions in the field."

We hope that these additions provide the required clarification.

Reviewer 1

To start with, we list all comments where we agree completely with reviewer 1, and where we have simply followed the recommendations without any comments or reservations:

Page 1, line 15: H+ → $H^+$
Page 1, line 15: Delete 'Consequently, ': The sentence is not a real consequence of the previous one (it's always true).
Page 1, line 18: Insert 'model' after 'HD-MINTEQ'
Page 3, line 1: Insert 'model as a basis' (or 'as a basis') after 'SHM'.
Page 4, line 18: It could/should be mentioned that in SMART/VSD the user can choose between a Gapon or Gaines-Thomas exchange model.
Page 5, line 31: Insert 'the' before 'UK'.
Page 5, line 32: Why 'however' after 'In one paper'? Suggest deleting!
Page 11, line 29: 'longest' → 'longer'.
Table 1: Caption: Maybe expand text? Write 'humic acids' and 'fulvic acids' instead of 'HA' and 'FA', resp., in the first column.
Table 2: Add 'scenario' after 'Background-acid-background' and after 'Background-salt' in the first line if the Table.
Figure 1, 2 and 3: Caption: (a) Insert 'in the four soil layers (O, E, B1, B2)' before 'as a function'; (b) insert 'deposition' before 'scenario'.
Figure 5: For 'aesthetic reasons' maybe interchange in the legends the lines 'SHM' and 'Ion exchange B' (twice) (?)

Below we provide a more detailed response to the other comments:

Page 6, line 1: What does the stand-alone '$K_i$' in eq.6 mean? In the next line it is referred to as 'Here the $K_i$ is …'?
ANSWER: This is a common way to clarify that there is an equilibrium constant that defines the relationship between the products and the reactants, but it is not an important clarification. However, since this is not really needed, we have decided to remove the K. The sentence that follows has been changed as a consequence.

Page 6, line 30: Eq.8 does not make sense: If I insert i=1,2,3,4 into it, I get 4 times the identical value for $logK_1$, …, $logK_4$, since the right-hand side of eq.8 does not depend on the index i!! Correct!
ANSWER: Thank you for pointing this out. Something had happened with an equation object that we had inserted, and therefore a term was missing. This has now been fixed.

Page 6, line 31: Same again!!

ANSWER: See above comment.

Page 8, line 2: There is no $Q_i$ in equations (10) or (11), thus one cannot solve for them!!
ANSWER: There is no error here - these equations should be seen as part of an equation system that is solved simultaneously. We have included the following sentence for clarification: "To calculate the partitioning of the reacting ions, Eqs. (10-12) are solved simultaneously as described by Kinniburgh et al. (1999)."

Page 11, line 6: It is not essential, but the scenario name is a bit 'strange'; and, actually slightly misleading, since the 3$^{rd}$ interval is not background depo (but close).
ANSWER: Valid point, we agree and have changed the scenario names as suggested by the reviewer

Page 11, line 8: Also this name is not 'unconfusing' … Why not just call the 'Acid' and 'Salt'? – just a thought …
ANSWER: See previous comment

Page 11, line 13: Should it be '(Gustafsson, 2016)', as in the References; or is the year there wrong?
ANSWER: The year given in the reference was wrong, it has now been changed to 2018

Page 12, line 2: Also in SMART/VSD $H^+$ exchange is modelled!
ANSWER: Correct, SMART/VSD has been added

Page 13, line 5: '(Gustafsson, 2016)'?
ANSWER: No, as written above the year is 2018

Page 13, line 29: 'SMART/VSD' could be added (?)
ANSWER: OK, it has been added. Similarly, we have also added SMART/VSD in the Abstract in two instances.

Page 14, line 12: The ending is a bit abrupt …
ANSWER: You are right, a Conclusions section is missing. We have now added a Conclusions section to the paper. It reads as follows:

"Ion-exchange equations, despite being relatively simplistic, predict the same type of response to changes in input chemistry as more advanced organic complexation models such as SHM, NICA-Donnan and WHAM. This is particularly true in cases when the pH variations with time are relatively small. If larger pH variations occur the differences in predicted pH and cation binding will be larger. The main reason for this is the acid-base chemistry of organic matter, for which acid dissociation occurs over a wide pH range. This effect is not captured correctly by the ion-exchange equations. For example, this may be an important point to consider for model initialization, as the soil water pH may have been much higher under preindustrial conditions than it was later during the acid rain era. The value of exchangeable $Ca^{2+}$ was found to be particularly sensitive in this regard. The method used to account for Al chemistry, i.e. whether or not letting the solubility of $Al^{3+}$ to be determined by a fixed gibbsite constant, was less important. Although a fixed gibbsite constant did cause stronger pH buffering than a more advanced model combining organic complexation and $Al(OH)_3$ precipitation, the effect was rather small. In summary, state-of-the-art organic complexation models should be preferred over ion-exchange equations for predicting pH and cation binding in dynamic acidification models, particularly if large pH variations occur during the simulated time period."

Page 15, line 24: Is it Gustafsson 2016 or 2018?
ANSWER: Gustafsson, 2018, as explained above

Figure 1, 2 and 3: The dotted red line is not visible in the graphs as such! Why not use solid lines with 4 (sufficiently) different colours?
ANSWER: Parts of the dotted lines do not appear as dotted (rather as solid) because of the very large number of points and because of the fact that there are frequent very small spikes in the simulation lines, we regret that. On the other hand, our reason for plotting some of the lines in rather similar colours is that it is then easy to quickly distinguish between the ion-exchange models (red-orange) and the organic complexation models (blue-green). Hence there is a slight conflict between the desire to quickly identify the exact data set and the desire to quickly distinguish the two groups of data. We have, for now, decided to keep the lines and colours unchanged.

Reviewer 2

First we would like to thank referee no. 2 for his or her interest in our manuscript and for constructive suggestions as outlined in the general comment. However, we have not made any changes in the manuscript. The reason for this is detailed below the referee comment, as follows:

"*This paper clearly fits within the scope of SOIL. However, it is a pity that it only includes hypothetical scenario analysis illustrating the differences between the two concepts (ion exchange' versus 'organic complexation' models). Overall, the paper is a rather technical description various*

*types of modelling approaches. This makes the paper less relevant for a broader audience. More important, however, a comparison with observations (such as pH, Al concentration and base saturation) is missing. Such a comparison is absolutely needed to judge the performance of both concepts. I realize that this is not an easy task, but a validation/application of the presented modelling concept by using e.g. the Gårdsjön observations or observations from Wesselink and Mulder (1995) and/or Bonten et al. (2001) must be doable. I therefore conclude that, although the current manuscript addresses a relevant issue and is well written, a major revision is required because a comparison with observations is missing."*

ANSWER: The purpose of the paper was "to review the process descriptions of currently used models as regards base cations and aluminium, and then to investigate the difference in model performance between the two types of model mentioned above, i.e. between ion-exchange models and organic complexation models". To this end, the paper was able to evaluate the response of the different exchange/complexation models to abrupt changes in chemistry. Because of this, the paper was also able to draw conclusions about the implications of choosing one exchange/complexation model over another, as well as the implications for the back-calculation of historical exchangeable pools (which remains uncertain in many of the current ecosystem biogeochemical models mentioned in the paper). The paper did not set out to "validate" biogeochemical models themselves, but it serves as a basis for future work devoted to the overall behaviour of biogeochemical models and how these are influenced by the choice of the investigated exchange/complexation models. For that eventual step, an evaluation of model performance compared to observations would certainly be relevant. See also reply to the topical editor above.

[revised manuscript text omitted]